# Effect of Ecological Water Supplement on Groundwater Restoration in the Yongding River Based on Multi-Model Linkage

Tian Nan [1,2] and Wengeng Cao [1,2,*]

1    The Institute of Hydrogeology and Environmental Geology, Chinese Academy of Geosciences, Shijiazhuang 050061, China
2    Key Laboratory of Groundwater Sciences and Engineering, Ministry of Natural Resources, Shijiazhuang 050061, China
*    Correspondence: caowengeng@mail.cgs.gov.cn

**Abstract:** Evaluating the effect of ecological water supplement on groundwater restoration quantitatively could produce positive contributions to both water cycle theory and surface–groundwater conjunctive management. Therefore, in this paper, a groundwater flow numerical model has been established after calculating the river section seepage rate using a fuzzy mathematical method in the Yongding River channel. The simulated results show that the model could accurately reflect the real groundwater dynamic features. Then, a data-driven random forest(RF) model has been established to quantitatively evaluate the contributions of the factors which influence the groundwater level variation. The Nash-Sutcliffe efficiency coefficient(NSE) of the RF model is 0.93. It shows excellent ability to identify the rising zone of groundwater level. The study shows that the infiltration capacity is strong in the upstream area of the Yongding River, and the seepage rate is over 0.7. The lowest seepage rate is 0.19 at the downstream end, while the seepage rate in the middle area is basically between 0.4 and 0.7. From 2018 to 2019, the ecological water supplement of the Yongding River has played a significant role in raising the groundwater level along the river channel. Additionally, its contribution analyzed by the RF model to the change of groundwater level is 25%. Groundwater exploitation is the most important variable affecting the groundwater level variation. The impact depth of groundwater level fluctuation reaches about 10 m. The impact range where the groundwater level average uplifts 1.86 m is 502.13 km$^2$. The influence direction gradually changes from around the ecological water supplement section to along the Yongding River channel. The groundwater level variation along the tangential direction of the Yongding River is slowing down. The groundwater level would entirely uplift with $170 \times 10^6$ m$^3$/year ecological water supplement of the Yongding River and $35.77 \times 10^6$ m$^3$/year groundwater mining reduction in the downstream area until 2035.

**Keywords:** Yongding River; ecological water supplement; fuzzy mathematics; groundwater numerical model; random forest



## 1. Introduction

For the sake of the sustainable development and utilization of groundwater resources, the MAR (managed aquifer recharge) technology system has been widely promoted and applied in recent years [1–4]. Artificial ecological water supplement for groundwater restoration in the North China Plain is an important practice of MAR technology at the watershed scale. Relative studies have shown that the river recharge could have a positive effect on the recovery of groundwater level within the local scale [5] but could also have a negative impact on groundwater quality [6]. On the one hand, river recharge increases the recharge pathway of groundwater aquifers, and then the amount of water recharged to the aquifers would be increased, and the water table would be uplifted [7,8]. On the other hand, due to the difference between the recharge source and groundwater, the variation of groundwater

quality would be present [9,10]. Some research has also shown that seasonal changes in surface–groundwater exchange would lead to ecological risks [11–13]. In summary, river ecological water replenishment can have a significant impact on the surface–groundwater circulation and ecological ambient. The influence of ecological water supplement on the groundwater level variation should be evaluated.

Numerical simulation is one of the most popular approaches for analyzing surface–groundwater interactions. Many kinds of numerical models have been proposed to study the surface–groundwater exchange. The mainstream models include the fully coupled model and semi-coupled model [14]. The fully coupled model requires detailed investigations of surface hydrological processes, water migration in the unsaturated zone and saturated zone, and then a reasonable conceptual model could be established [15–17]. The fully coupled model is advantageous for small-scale studies. However, for a medium to large regional scale study, the dimension explosion of parameters and the computing cost would create crucial problems [18–20]. The semi-coupled model based on MODFLOW can deal with surface water transport process as boundary conditions. Additionally, the modeling process is simpler and more flexible than the fully coupled model. It is often used in the study of surface–groundwater interactions at the regional scale [21]. However, whether it is fully coupled or semi-coupled, the hydrogeological parameters in the surface–groundwater conversion process are the key factors affecting the model accuracy [22–24]. In addition, with the development of computer science, because of the high simulation precision and the flexible data processing flow, data-driven models have been widely used in groundwater hydrogeological research. Various machine learning models have been widely applied to predict the variation trend of groundwater level and budget [25–27]. Within these machine learning algorithms, the random forest algorithm has flexible parameter adjustment, high prediction accuracy, strong generalization ability, and is not likely to produce an over-fitting problem. It has fewer requirements on data quality and has strong robustness to feature variable selection. At present, it is commonly used for solving both classification and regression problems. Moreover, the random forest model is also friendly for the explanation of feature variables [28–30].

The Yongding River flows through Beijing, Hebei, and Tianjin and into the Bohai Sea. It has begun to dry up since the 1980s. By 2010, the Yongding River had actually been cut off because of the overexploitation in the upstream areas [31,32]. In an effort to recover the ecological environment and the socio-economic vitality around the Yongding River, various water supplement projects have been launched, including reservoir water replenishment, reclaimed water replenishment, the Yellow River Diversion Project, and the South-to-North Water Diversion Project [33,34]. Since 2017, the amount of water recharged from these projects mentioned above has gradually increased [35]. The treatment effect of ecological water supplement on groundwater aquifers along the Yongding River region needs to be evaluated. However, the research on water supplement of the Yongding River is mainly focused on the influence on groundwater quality in the river channel [36–38], but the systematic and comprehensive research on the relationship between ecological water supplement and groundwater level variation has not been formed, and the contribution degree to water level recovery and the influence range of water supplement are not clearly evaluated [39,40].

Therefore, this study has proposed a systematic method based on multi-models for effective evaluation of the ecological water recharge effect on groundwater restoration. First, a fuzzy mathematical method has been used to figure out the seepage rate along the Yongding River channel. Additionally, the ecological water infiltration amount of each section along the Yongding River has been calculated. Then, a numerical simulation model under ecological water supplement conditions has been established for analyzing the regional groundwater level and budget variation. Additionally, the affected area of ecological recharge has been determined. Finally, the contribution of the ecological water recharge for the recovery of groundwater level has been quantitatively evaluated by using the random forest model.

## 2. Materials and Methods

### 2.1. Geological and Hydrogeological Conditions in the Study Area

The study area is from Mentougou to Beichen along the main Yongding River channel. The total study area is 3842.05 km² (Figure 1a). In addition to urban areas, there are farmlands, forestlands, and grasslands in the study area [41,42]. The length of the Yongding River channel in this study is 162.5 km.

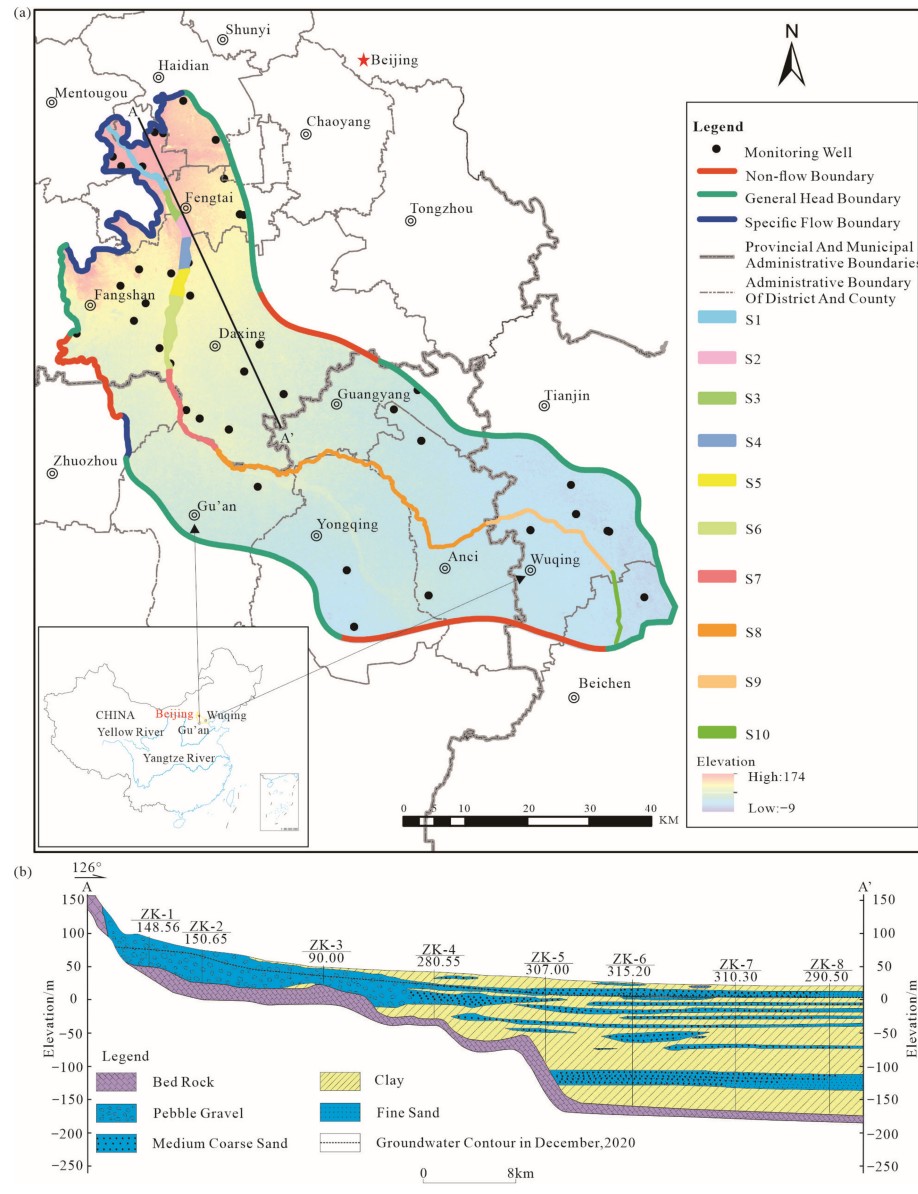

**Figure 1.** (**a**) location of study area; (**b**) hydrogeological profile in study area (modified after Hao et al. (2014)) [5].

The study area has a typical monsoon climate. The average intensity of precipitation is 595 mm/year, and the average intensity of water surface evaporation is 1120 mm/year (1959–2019). Precipitation is mostly concentrated from June to September each year [43]. The Yongding River Comprehensive Management and Ecological Restoration Project has been in operation for more than 2 years [44,45], and the impact on the groundwater level fluctuation has been gradually revealed. The regional groundwater flows from the northwest to the southeast.

The aquifers mainly consist of coarse-grained sediments in the piedmont area, and gradually changed into finer-grained sediments in the downstream plain (Figure 1b) [46]. The

piedmont area is mainly from Mentougou to Fengtai. In the piedmont area, the aquifer is mainly a single phreatic aquifer which is composed of relatively coarse gravel. The thickness of the phreatic aquifer is about 50 to 200 m. The range of hydraulic conductivity changes from 10 to 200 m/d. In the middle reaches of the Yongding River, the aquifers have changed into a multi-layer structure. The aquifers are mainly composed of middle and fine sand, with a few gravels. The thickness of aquifers is about 100 to 150 m. Additionally, the hydraulic conductivity in the middle reaches is usually within 30 m/d. The downstream area is roughly from the Yongqing River channel to the very southeast part of the study area. The aquifers are composed of fine sand. Each aquifer is relatively thin, and the thickness is usually less than 10 m. Additionally, between aquifers, the area is filled by a large number of clay layers. Therefore, the hydraulic conductivity in the downstream aquifers is generally less than 10 m/d [47,48].

The main recharge items in this area include mountain lateral runoff, precipitation infiltration, and water infiltration of the Yongding River; the main discharge items include artificial exploitation, evaporation, and lateral outflow [49]. The bottom boundary of the Quaternary aquifer is impermeable sediment [50].

### 2.2. Workflow and Data Collection

The flowchart of this research is shown in Figure 2. Firstly, the relevant data will be collected and preprocessed. After that, the seepage rate of each section of the Yongding River channel will be calculated using the fuzzy mathematics method. Then, the amount of infiltration can be estimated. Next, using the calculation result of ecological water supplement infiltration, a loosely coupled numerical model will be established. Groundwater restoration effect will be analyzed by using the numerical model. Finally, a random forest model will be used to evaluate the contributions of source and sink items on the groundwater level variation. The specific data and methodology descriptions have been shown as follows.

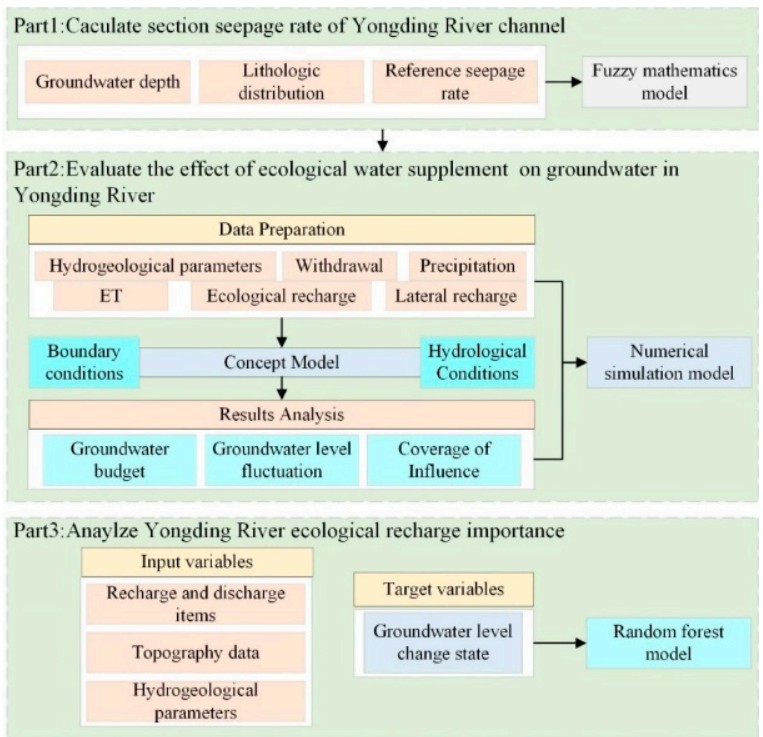

**Figure 2.** Flowchart of this study.

The data used for the study consist mainly of precipitation, evaporation, groundwater extraction, topography, water levels, and river recharge. Precipitation and evaporation data come from the monthly observation data of the National Meteorological

Administration. Topographic data come from Alos satellite remote sensing interpretation results [51]. Groundwater exploitation data are collected from Beijing, Tianjin, and Hebei hydrological reports [52–57]. The Yongding River ecological water supplement data from 2018 to 2019 are collected from Haihe River Water Conservancy Commission [58,59]. The water level data are from monthly observations from national groundwater monitoring wells in the area from 2016 to 2019 [60]. The drilling data of the sections are derived from the hydrogeological survey of the Institute of Hydrogeology and Environmental Geology, Chinese Academy of Geological Sciences, in 2019 [61].

*2.3. Models*

2.3.1. Fuzzy Mathematics Model

Fuzzy mathematical methods are used to solve for the river section seepage rates. The method mainly satisfies the mathematical calculation by constructing an affiliation function to quantify the qualitative indicators [62]. Since the Yongding River channel recharges the shallow groundwater along its route in the form of seepage, and the seepage rate is mainly determined by the lithology of unsaturated zone [63], this study establishes an affiliation function between the lithology distribution of the unsaturated zone and the seepage rate. The infiltration test data of the Yongding River channel, provided by the Haihe Water Conservancy Commission, are used as the reference value to calculate the permeability of each section of the Yongding River channel. Firstly, the river channel is divided into four types of lithology: clay and silt, fine sand, medium coarse sand, and pebble gravel (Figure 1b). The empirical values of the seepage rate of the four types of lithology are given, and the permeability evaluation values of the four types of lithology are obtained after standardization $S_i$ ($i = 1, 2, 3, 4$; $S_i \in A$) ; the Yongding River is divided into n sections according to the recharge conditions. According to the area $A_{ni}$ of four types of lithology in the profile and the total area $A_n$ of the section, the proportion $P_{ni}$ of four types of rock strata in the $n$th section is calculated:

$$P_{ni} = \frac{A_{ni}}{\sum A_n} \tag{1}$$

Then, the combined permeability evaluation value of the $n$th rock section is calculated as $S_n$ :

$$S_n = \sum P_{ni} S_i \tag{2}$$

Based on the maximum and minimum seepage rates $\lambda_{max}$, $\lambda_{min}$ gained from the infiltration test, a linear equation between the seepage rate of the section and the permeability evaluation value $S_n$ is developed:

$$\lambda_n = \frac{S_n(\lambda_{max} - \lambda_{min})}{S_{nmax} - S_{nmin}} + \lambda_{min} \tag{3}$$

Solving the equation can obtain the seepage rate of each section.

2.3.2. Numerical Simulation Model

The numerical simulation model is the main method applied to estimate the effect of ecological water supplement on the groundwater level and budget variation.

The groundwater aquifers in the Yongding River region were generalized as a three-dimensional heterogeneous transient groundwater flow system. The model was composed of two conceptual aquifers. The upper layer was the mainly concerned unconfined aquifer group, and the lower layer was the confined aquifer group. The western boundary receiving the piedmont lateral recharge was set as a specific flow boundary, and the southern and northern parts were set as a no-flow boundary. The remaining boundaries had the variation water exchange with the outside area, and then they were set to the general head boundary. Precipitation, river recharge, and irrigation infiltration were designed as recharge boundaries. The model bottom boundary had been regarded as a no-flow boundary. The main

hydrogeological parameters used in the model were hydraulic conductivity, specific yield, and storage coefficient. The distribution of the initial hydrogeological parameters were assigned based on the interpretation of the hydrogeological test, drilling work, and the previous research results in the same study area [64,65]. These parameter values were finally determined during the model calibration process.

The model was developed with MODFLOW. The grid size was 500 m × 500 m. The first layer was discretized into 15,850 effective cells, and the second layer was discretized into 11,653 effective cells. The model was discretized into 48 stress periods from January 2016 to December 2019. In the simulation period, ecological water supplement was allocated from April to July.

The main sources of ecological water supplement include the discharge of water from the Guanting reservoir and the water introduced from the South-to-North Water Diversion Project. Because of the shallow groundwater level, the surface water generally has no direct contacts in the study area. The water first enters the Yongding River, and then infiltrates into the unsaturated zone. After partial interception, the rest of it is finally recharged to the shallow aquifers. Additionally, the amount of infiltration has been calculated with the fuzzy mathematics method proposed in this paper.

The distribution of precipitation recharge rates was figured out by the amount of precipitation and infiltration coefficient. Evapotranspiration was calculated by the Evapotranspiration Package. Because the mathematical relationship between evaporation and groundwater level was clear, the groundwater evaporation distribution was the same as the area where the groundwater depth was less than 4 m [66]. Similarly, the evaporation intensity was verified by model fitting. Artificial exploitation of groundwater was put into the model in the form of mining well and calculated by well module.

The trial-and-error method was applied for model identification and verification. The groundwater flow field measured on 31 December 2019 and the observation groundwater level process line were used to identify and verify the model simulation results.

### 2.3.3. Machine Learning Model

The random forest selected in this study is a popular machine learning strategy (generated in Bell Labs in the 1990s), which belongs to the ensemble learning method in machine learning algorithms. It works by generating multiple classifiers, each of which has an independent learning and predicting ability. These predictions of the classifiers are finally combined into an integrated result. Additionally, the accuracy of the final result is much better than any single prediction result. At present, it is commonly used to simulate and analyze the evolution of groundwater level and water quality [67,68].

In this study, whether the water level rises was taken as the target label (rising to 1, not rising to 0), and the groundwater source and sink items, hydrogeological parameters, and topographic features were taken as the feature variables. The 15378 samples' data in the study area were randomly divided into 80% training data and 20% testing data. The Random Forest Classifier package in the Python ensemble library was used to establish the model.

### 2.3.4. Model Performance Evaluation Metrics

The Nash–Sutcliffe efficiency coefficient (*NSE*), the model sensitivity, and the specificity are used as the evaluation criteria. *NSE* is used to evaluate the accuracy of the regression models. The sensitivity evaluates the accuracy of positive samples prediction. The specificity reflects the accuracy of negative samples prediction. The criterion is close to 1, illustrating that the model is more reliable.

$$NSE = 1 - \frac{\sum_{i=1}^{n} (o_i - y_i)^2}{\sum_{i=1}^{n} (o_i - \bar{o}_i)^2} \tag{4}$$

$$\text{Sensitivity} = \text{TP}/(\text{TP} + \text{FN}) \tag{5}$$

$$\text{Specificity} = \text{TN}/(\text{TN} + \text{FP}) \tag{6}$$

where $y_i$ is the simulated value, $o_i$ is the observed value, and $\bar{o}_i$ is the average value of the observed values. $i$ marks the number of observation wells used in the model performance evaluation and $n$ is the total number of observation wells. TP is the positive sample that the model predicts correctly, FN is the negative sample that the model predicts incorrectly, and the sum of TP and FN is the total number of correct positive samples. TN is the negative sample which has been correctly predicted by the model, FP is the positive sample which has not been correctly predicted by the model, and the sum of TN and FP is the total number of correct negative samples. TP, FN, TN, and FP values are obtained from the simulation results.

The influence of feature variables on water level change is evaluated by feature importance value distribution and Shapley value distribution [69]. The feature importance distribution is applied to assess the contribution of feature variables to water level variation by using the Gini impurity reduction method. The Gini impurity is used to decide the splitting direction of the decision trees. Disturb the order of feature variables separately, and the change of the orders will influence the accuracy of the model. The more important the feature variable is, the more decreases of the accuracy of the model will be produced. Shapley value is used to calculate the marginal contribution of every feature variable. Corresponding to each sample, the random forest model will generate a prediction value, which is the Shapley value. The Shapley value not only provides the degree of the feature variable influence, but also reflects the positive and negative influence of the feature variables. In this study, the greater the Shapley value is, the more the feature variable promotes the rise of water level, and vice versa.

## 3. Results

### 3.1. Evaluation of the Infiltration Volume of the Yongding River Channel

The seepage rate of each water supplement section of the Yongding River (Figure 1a) calculated by fuzzy mathematics method is shown in Table 1. Then, as the ecological water replenishment volume data in each section of the Yongding River channel have been determined, the infiltration amount of each section is calculated.

**Table 1.** Ecological water supplement infiltration volume and seepage rate for each section of the Yongding River.

| Section | Seepage Rate | Ecological Water Supplement ($\times 10^6$ m$^3$) | Infiltration Volume ($\times 10^6$ m$^3$) |
|---|---|---|---|
| S1 | 0.72 | 4.90 | 3.54 |
| S2 | 0.72 | 44.07 | 31.82 |
| S3 | 0.76 | 3.66 | 2.78 |
| S4 | 0.62 | 40.52 | 25.02 |
| S5 | 0.43 | 10.52 | 4.50 |
| S6 | 0.38 | 22.83 | 8.68 |
| S7 | 0.48 | 12.05 | 5.72 |
| S8 | 0.48 | 0 | 0 |
| S9 | 0.22 | 0 | 0 |
| S10 | 0.19 | 0 | 0 |
| Total | 0.59 | 138.55 | 82.06 |

The lithology of section S3 in the upstream of the Yongding River is mainly composed of medium coarse sand and coarser grained sediment, and at the same time, the proportion of aquifer is much higher than the aquitard. Therefore, the seepage rate in this section is the highest in all sections. It reaches 0.76 in section S3. Moreover, because the aquifer of S1 and S2 is slightly thinner than S3, the seepage rates of S1 and S2 are also kind of lower than S3, but the values are also over 0.7. In the lower reaches of S10, due to the high proportion of clay, and the aquifers mainly composed of fine sand, the seepage rate is the lowest in S10 of 0.19. In other sections, the clay content gradually rises from upstream to downstream

areas. Additionally, the sediment constructing the aquifers has changed from coarse to fine. Then, the seepage rate distribution shows a synchronous and gradually decreasing trend. The results of this study are highly in agreement with Ji's study which used the numerical simulation method in 2021. However, the method proposed in this paper is more flexible and convenient.

The infiltration amount in each section of the Yongding River is calculated with the seepage rate. The water recharge from S2 is the largest in all sections, which is up to $31.82 \times 10^6$ m$^3$, accounting for 38.8% of the total ecological water supplement. The ecological water recharge from S3 is the smallest one, about $2.78 \times 10^6$ m$^3$, accounting for only 3.4% of the total ecological water supplement. About 60% of the ecological water supplement volume in the study area could infiltrate and recharge the shallow aquifer. The loss of the supplement water is partly intercepted by the vadose zone, and partly consumed in the process of water transport through evapotranspiration. Substituting the calculation results of the ecological water supplement infiltration volume into the numerical model, the coupled model could be used to analyze the effect of the ecological water supplement of the Yongding River on groundwater flow system.

*3.2. Numerical Model Performance*

The numerical model has been fitted and calibrated by comparing the observation data of 64 typical observation wells in the study area from 2016 to 2019 and the measured groundwater flow field on 31 December 2019 with the simulation value.

The identification and verification results show (Figure 3) that the simulated flow field fits well with the measured flow field. There is a slight difference between the southern and eastern boundaries. At the same time, the simulated groundwater levels have a similar variation trend with that in the observation wells. It proves that the model could precisely describe the groundwater variation features through the ecological water supplement period. The mean absolute error between the measured data and the simulated data of groundwater level of all the observation wells in the calibration period is 0.91 m. As shown in Figure 3b, all the simulated and observed fitting points are very close to the 1:1 line, and the *NSE* is 0.82. This means that the simulation result has an overall good performance. Local error mainly comes from insufficient knowledge on the hydraulic parameters and regional boundary conditions. On the whole, the model established in this paper could correctly present the dynamic features of groundwater flow in the Yongding River channel and could be used to quantitatively analyze the spatial and temporal effects of ecological water supplement on groundwater flow system.

With the model corrected by fitting the groundwater flow field and typical observation wells, the location of flow boundary has been determined. The lateral groundwater flow mainly comes from the southern part of Haidian and Mentougou. It is the lateral recharge of the study area. The total amount of the lateral recharge was $14 \times 10^6$ m$^3$/year before 2017, and it increased to $100 \times 10^6$ m$^3$/year after 2017. Then, the no-flow boundary and general head boundary have been calibrated (Figure 1a). Finally, the amount of outflow through the boundary has been figured out. According to the change of groundwater flow field in 2018 and 2019, the groundwater mainly flows out from the eastern and southern parts of the study area. The flow-out boundary is mainly from east Fengtai. Then, the northern part of Wuqing and the southern part of Beichen are also the flow-out boundary. The calibrated model has captured the boundary features. The out flow calculated by the model was $263 \times 10^6$ m$^3$/year before 2017, and it decreased to $40 \times 10^6$ m$^3$/year after 2017.

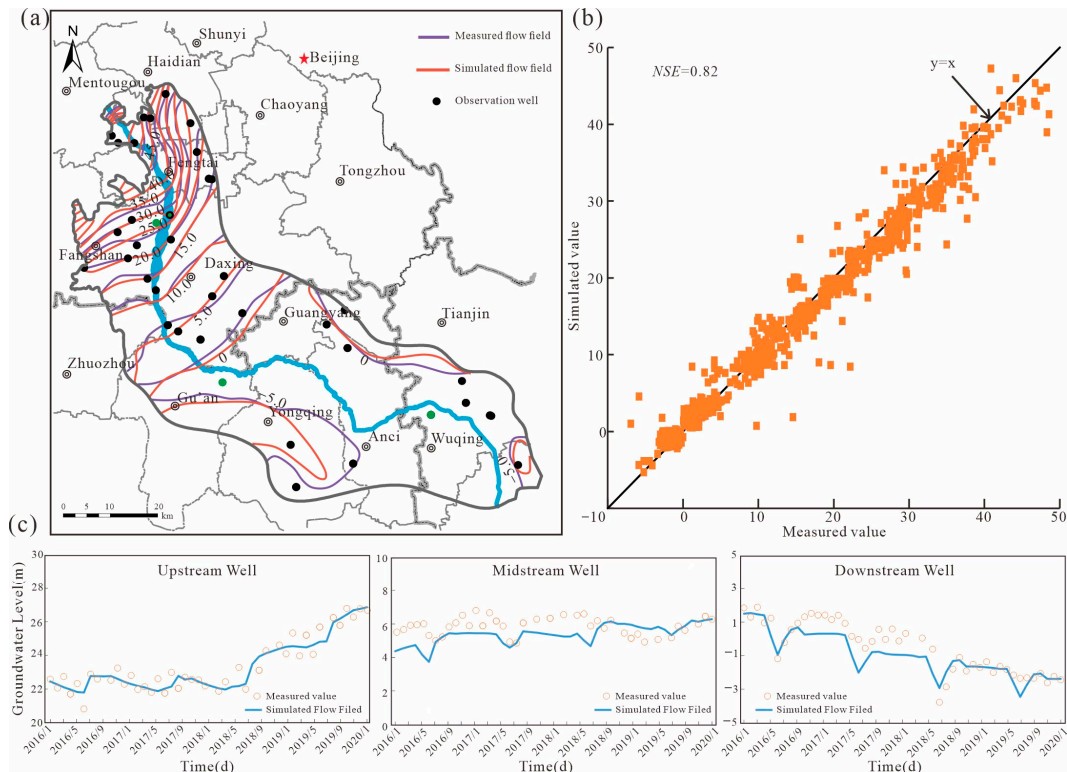

**Figure 3.** (**a**) the fitting plot at the end of the simulation period; (**b**) the fitting curve of all simulated and measured groundwater levels; (**c**) the fitting curve of typical observation well in the ecological water supplement zone.

The hydraulic conductivity and specific yield are the most important parameters to control groundwater flow. The hydraulic conductivity presents the capacity of aquifers to transport the groundwater based on the difference of hydraulic gradient. The specific yield reflects the water release ability of the unconfined aquifer. The smaller the specific yield is, the more obvious the groundwater level fluctuation is. Figure 4 shows the calibrated distributions of hydraulic conductivity and specific yield. The maximum calibrated hydraulic conductivity value is about 150 m/d, and it is presented in the southern part of the Mentougou area which is close to the mountain area. The aquifer is a single structure composed of gravel. Because the aquifers change into multi-layer structure and the sediments change into middle to fine sand, the calibrated hydraulic conductivity decreases. To the east of the high values area, the hydraulic conductivity values drop to below 50 m/d. The hydraulic conductivity calibrated in the middle part of the study area is usually around 20 to 30 m/d. In the downstream of the Yongding River, the hydraulic conductivity calibrated in this model changes from 15 to 5 m/d. The hydraulic conductivity distribution is in accord with the sedimentary law in the study area. Additionally, the calibrated specific yield distribution has a similar variation trend with the hydraulic conductivity distribution. The value of specific yield is relatively high in the piedmont area, which can reach 0.18. Then, the calibrated specific yield value reduces to 0.15 in the south of the Daxing area which is the middle part of the study area. The smallest calibrated value of specific yield is 0.07 in the downstream of the Yonding River. All of the calibrated parameter results in this paper are highly consistent with those of previous studies [5,70], but the study area has been further extended compared with the former studies.

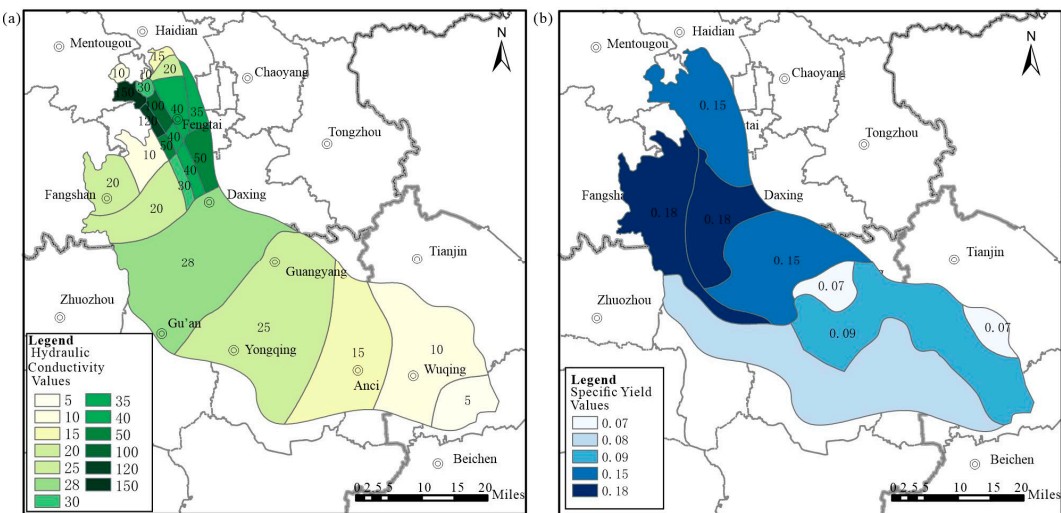

**Figure 4.** (**a**) the identification verified hydraulic conductivity distribution; (**b**) the identification verified distribution.

*3.3. Influence of the Ecological Water Supplement of the Yongding River on the Groundwater Flow Systems*

3.3.1. Actual Groundwater Level Variation in the Study Area from 2017 to 2019

Comparing the groundwater flow field in the study area between 2017 and 2019, the groundwater level was raised in the upstream area where the ecological water supplement had arrived (Figure 5). The groundwater level recovery range was generally between 2 and 8 m. The groundwater level in the piedmont generally increased by more than 8 m. In the area between Fangshan and Daxing, the river channel section could only get a few infiltrations of ecological water supplement. Thus, the groundwater level showed a small variation. The groundwater changed commonly between −2 m and 2 m. The groundwater level decrease mainly occurs in the southeastern parts of the study area. The groundwater level in these areas generally decreased by about 2 m. One reason for the groundwater level decline was that the ecological water supplement has not arrived at the downstream area from 2018 to 2019. The other reason was that the groundwater exploitation in the downstream area exceeded the natural recharge.

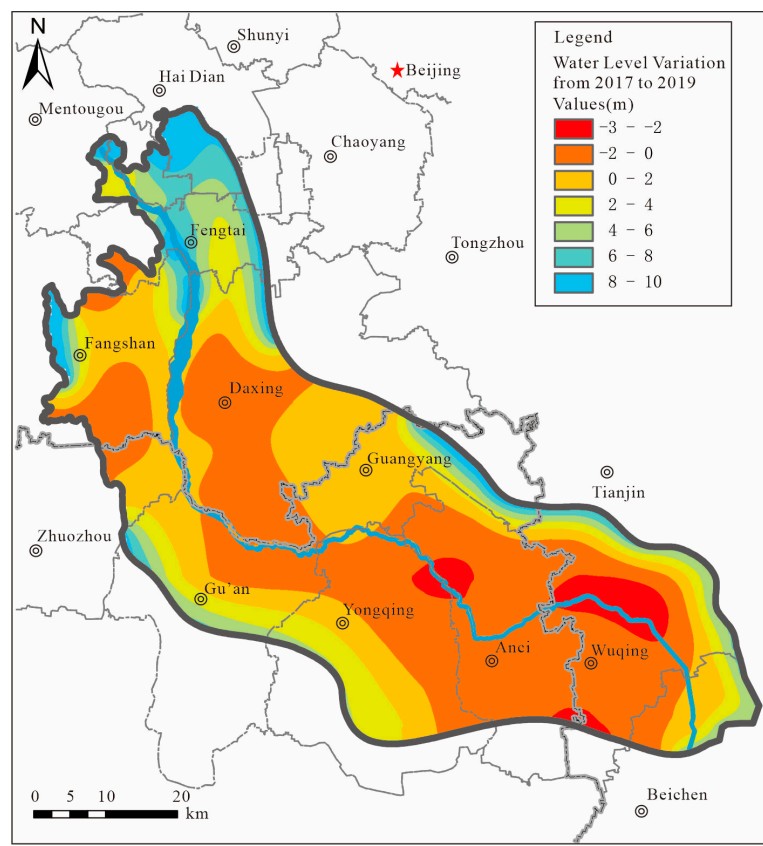

**Figure 5.** Differences in groundwater flow fields in the study area before and after ecological water supplement.

### 3.3.2. Groundwater Budget Variation before and after Ecological Water Supplement

It can be found from the change of groundwater budget (Table 2) that the ecological water recharge of the Yongding River could increase the recharge of $82 \times 10^6$ m³/year for the study area, accounting for 9.47% of the total recharge. After ecological water supplement, the shallow groundwater changed from negative equilibrium state to positive equilibrium state. The equilibrium difference was $383 \times 10^6$ m³/year, and the contribution proportion of ecological water supplement infiltration was 21.41%. The effect of ecological water supplement on groundwater quantity change was significant in the study area.

**Table 2.** Groundwater budget before and after ecological water supplement in the study area from 2016 to 2019.

| | | Before Ecological Water Recharge | | After Ecological Water Recharge | |
|---|---|---|---|---|---|
| | Budget Tems | Volume (×10⁶ m³/year) | Percentage | Volume (×10⁶ m³/year) | Percentage |
| Recharge | Precipitation infiltration | 578 | 68.65% | 507 | 58.55% |
| | Mountain front recharge | 141 | 16.75% | 98 | 11.32% |
| | Lateral inflow | 14 | 1.66% | 100 | 11.55% |
| | Ecological water supplement infiltration | 0 | 0.00% | 82 | 9.47% |
| | Regression quantity of well irrigation | 109 | 12.95% | 79 | 9.12% |
| | Total recharge | 842 | 100% | 866 | 100.00% |

**Table 2.** *Cont.*

| | | Before Ecological Water Recharge | | After Ecological Water Recharge | |
|---|---|---|---|---|---|
| | **Budget Tems** | **Volume (×10⁶ m³/year)** | **Percentage** | **Volume (×10⁶ m³/year)** | **Percentage** |
| Discharge | Evaporation | 18 | 1.80% | 3 | 0.46% |
| | Lateral outflow | 263 | 26.10% | 40 | 6.19% |
| | Exploitation | 725 | 72.10% | 603 | 93.34% |
| | Total discharge | 1006 | 100.00% | 646 | 100.00% |
| | Recharge and discharge difference | −163 | | 220 | |

### 3.3.3. The Effect of the Ecological Water Supplement of the Yongding River on Groundwater Level Variation

For further explaining the effect of ecological water supplement on groundwater level variation in the study area, the single factor variation scenario has been simulated. The simulation results of the model with ecological water supplement conditions and the model without ecological water supplement conditions were compared. The simulated flow fields on 31 December 2019 of the two models have been compared with the uplift variation variogram (Figure 6a). It can be found that the groundwater level in the upstream area which received sufficient ecological water recharge has obviously increased by over 4 m. Additionally, the maximum increase differences in the groundwater level variation with the two different conditions are located in the upstream area of the Yongding River nearby Fengtai and Daxing. The maximum increase in groundwater level is 8 to 10 m. The groundwater level rising zone could extend to the northern part of Gu'an. Additionally, the average groundwater level increase is 1.86 m.

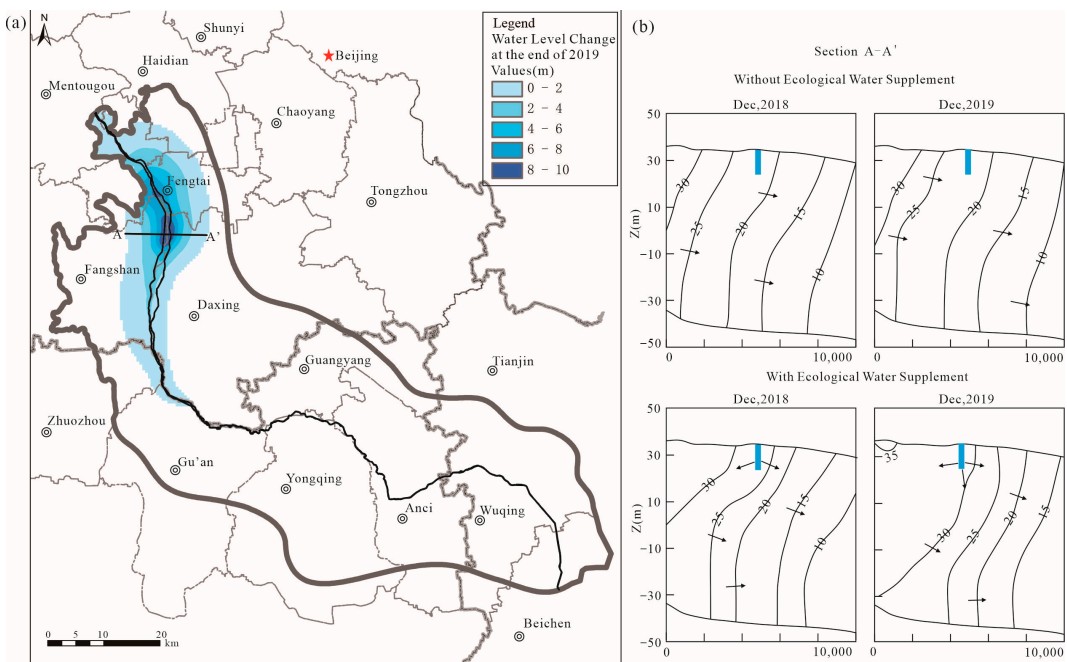

**Figure 6.** (**a**) variation in groundwater level between with and without the ecological water recharge at the end of 2019; (**b**) variation in cross section groundwater level between with and without the ecological water scenarios from 2018 to 2019.

A typical cross section has been selected to demonstrate the differences in flow field after the ecological water supplement. The selected cross section is located in the middle of S4 where the groundwater level changes the most obviously. Through the comparison of the cross-section flow fields, it can be seen (Figure 6b) that the groundwater level

and hydraulic gradient decreased from west to east along with decreasing land surface elevations under the without ecological water recharge condition. Additionally, the flow trend has hardly changed from 2018 to 2019. Under the ecological water supplement condition, the groundwater field is different from that under the without ecological water supplement condition, and the groundwater level obviously increased. The ecological water recharge caused the groundwater level near the river channel to rise significantly in 2018, and the average groundwater level increase could be about 5 m. As the ecological water supplement progresses, the groundwater level gradually increased up to around 10 m. The influenced range from the center of the Yongding River channel had gradually spread to more than 5 km near the Yongding River channel by 2019.

### 3.3.4. The Impact Range of the Yongding River Ecological Water Supplement

Comparing the flow field between with ecological water supplement and without ecological water supplement (Figure 7), it can be seen that with the supplement water gradually recharged into the aquifers, the groundwater level on both sides of the Yongding River has increased. Taking the interpolation errors of the model and the minor fluctuations in the groundwater level into account, the groundwater level increase greater than 0.5 m has been selected as the standard to determine the impact range.

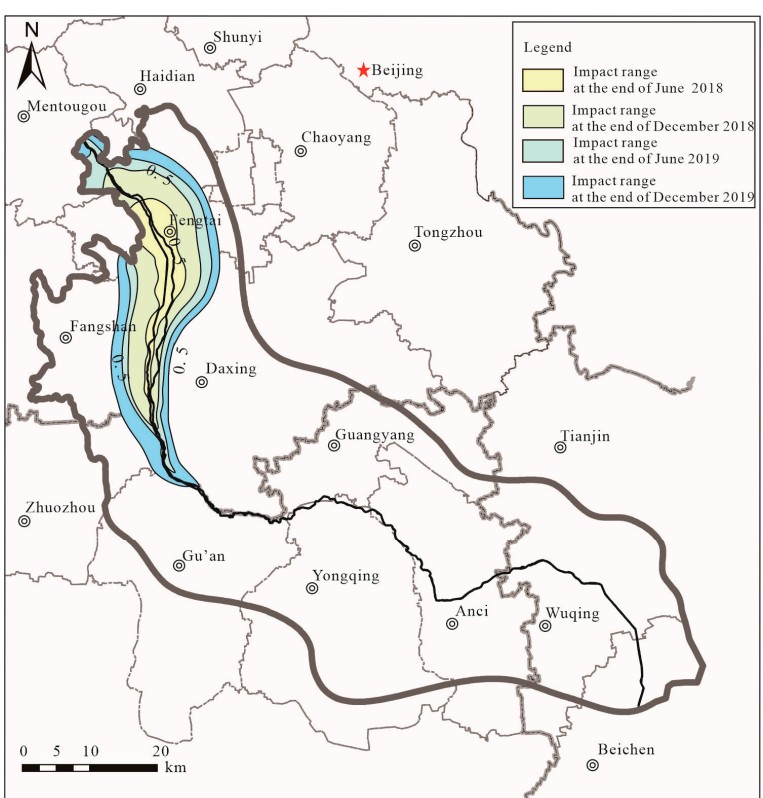

**Figure 7.** Variogram of impact range of the ecological water supplement of the Yongding River.

By the end of 2018, the farthest impact range of ecological water supplement could reach Zhuozhou. At this time, the farthest influence distance was 6.18 km, and the total impact range was 246.15 km$^2$. From the end of 2018 to the beginning of 2019, the impact range of ecological water supplement in the study area continued to expand, and the distribution feature of impact range gradually changed from developing along the Yongding River channel to the planar development pattern. The main development direction was northwest–southeast, which was consistent with the direction of groundwater flow. By 2019, the groundwater affected by ecological water supplement had reached Gu'an as far as possible. The areas affected by ecological water recharge include Beijing urban area, Fangshan District,

Daxing District, Zhuozhou City, and Gu'an County. The farthest influence distance at the end of the year reached 8.13 km, and the total impact range was 502.13 km$^2$. In addition, it can be found that when the maximum influence distance exceeds 6 km, the tangential impact range development rate along the river channel gradually decreases, and the development rate of the impact range along the river channel increases.

## 4. Discussion

### 4.1. Contribution of the Ecological Water Supplement in the Yongding River Channel to Groundwater Restoration

The random forest model can be used to identify the groundwater level uplift zones, and then the feature importance value and Shapley value can be used to estimate the contribution of the feature variables to the groundwater level restoration.

The selected feature variables (Table 3) include meteorological indicators, landform indicators, human activity indicators, and hydraulic features.

**Table 3.** Dataset of the driving factors.

| Feature Variables | | | |
|---|---|---|---|
| **Type** | **Indicator** | **Source** | **Form** |
| Meteorological factors | Precipitation | Statistical data | Accumulate |
| | Evaporation | Statistical data | Accumulate |
| Topographical factors | Landform | Remote sensing data | Distributed |
| | Lateral inflow | Numetical model | Distributed |
| Human factors | Ecological water supplement | Statistics caculating | Accumulate |
| | Artificial mining | Statistical data | Accumulate |
| Hydraulic feature | Hydraulic conductivity | Numerical model calibration | Distributed |
| | Specific yield | Numerical model calibration | Distributed |
| **Labeled data** | | | |
| Groundwater level rises or not | | Yes (1) | |
| | | No (0) | |

The *NSE* of the established random forest model is 0.93, the sensitivity is 0.92, and the specificity is 0.94, indicating that the model can well reflect the influence of feature variables on the groundwater level variation in the study area.

As can be seen from the distribution of feature importance value (Figure 8), human activities are the main driving factors affecting groundwater level recovery in the study area, with the contribution of 58%. According to the Shapley (Figure 9) distribution map, it can be seen that artificial groundwater mining mainly restrains the rise of groundwater level, with the contribution of 33%. The greater the amount of exploitation is, the more difficult it is for the water level to rise. The long-time over-exploitation is also the main reason leading to the regional groundwater level drawdown in the last decades [71].

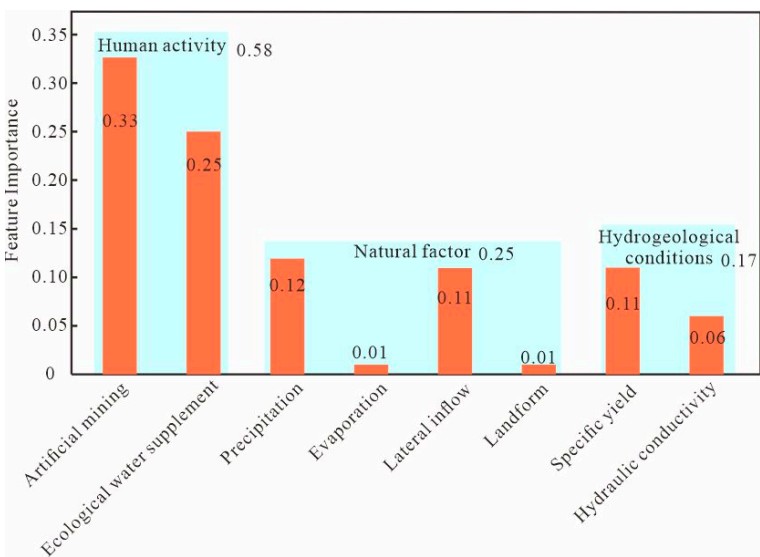

**Figure 8.** Feature importance map of driving factors.

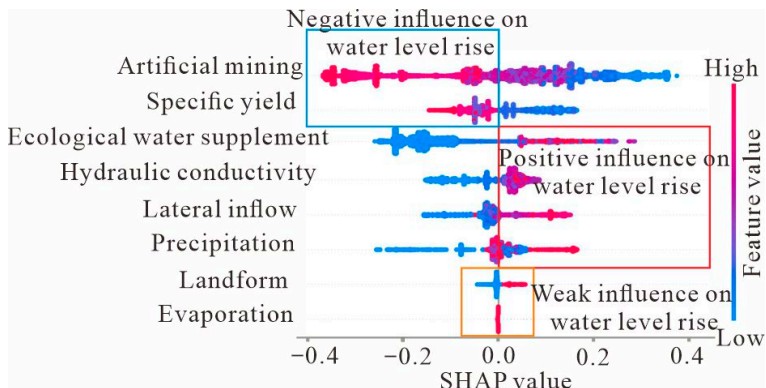

**Figure 9.** Shapley value distribution of driving factors.

The ecological water supplement infiltration could play a significant role in promoting the recovery of groundwater level, and the contribution is 25%. With the increase in ecological water supplement infiltration, and especially because of the great infiltration capacity in the upstream area of the Yongding River, the groundwater level in the study area has increased obviously [72]. The ecological water recharge in the groundwater cycle will become more and more important with sufficient surface water and reclaimed water sources.

Secondly, precipitation would have a positive effect on the recovery of groundwater level in the study area, with the contribution of 12%. The precipitation recharges the groundwater in the form of areal supply [73]. The precipitation in the study area is the steady recharge source. It is the most important recharge item in the study area.

The lateral recharge and specific yield contributions are more than 10%. The lateral recharge usually depends on the hydraulic gradient and the lithology of calculation section. The greater the hydraulic gradient and permeability are, the greater the lateral recharge is, and the more the groundwater level can be uplifted [74]. In the study area, the piedmont area has the largest lateral recharge, but the recharge section is relatively short, and the total lateral recharge in the study area is small. Therefore, the lateral recharge cannot be the main recharge source in the study area.

The specific yield reflects the groundwater level change ability in the study area [75]. It can be seen that the larger the specific yield is, the less obvious the groundwater level change is. The groundwater aquifers in the upstream area of the Yongding River are mainly composed of coarse sediments. The specific yield in the upstream area is large, and the groundwater level

fluctuation is significant. However, in the middle and low reaches of the Yongding River, the groundwater aquifers are composed mainly of fine sand. The groundwater level variation impacted by the specific yield is not obvious.

The contributions of other feature factors to the groundwater level variation in the study area are very small.

### 4.2. Suggestions of Ecological Water Supplement and Groundwater Exploitation

Using the established groundwater flow numerical model in this study, the prediction study was carried out for the purpose of achieving the groundwater level rise in the entire area in 2035. By adjusting the amount of ecological water supplement and groundwater exploitation, in the study area, the groundwater level could be entirely recovered state. Additionally, the specific scheme proposed in this study is shown as follows:

1. The ecological water supplement should increase by $40 \times 10^6$ m$^3$/year in the downstream area. The potential artificial ecological water supply sources mainly include the water from the mid-route of the South-to-North Water Diversion Project, the water from the Guanting Reservoir, and the reclaimed water collected from the cities along the Yongding River. Then, the total ecological water supplement in the study area is $170 \times 10^6$ m$^3$/year.
2. The groundwater exploitation should be reduced by $35.77 \times 10^6$ m$^3$/year in the downstream area, including the eastern part of Yongqing County, the southern part of Langfang City, the southern part of Wuqing District, and the western part of Beichen District.

Figure 10 shows the groundwater level variation from 2017 to 2035 under the scheme mentioned above. In the upstream area, the groundwater level would increase by over 20 m because the infiltration of ecological water supplement is far more than other sections. By contrast, the filtration of ecological water supplement is very limited in the downstream area, and the groundwater level just increases by about less than 0.3 m/year. Between the upstream area and downstream area, the groundwater level could increase by 2 to 8 m, and the average groundwater level uplift is around 5 m.

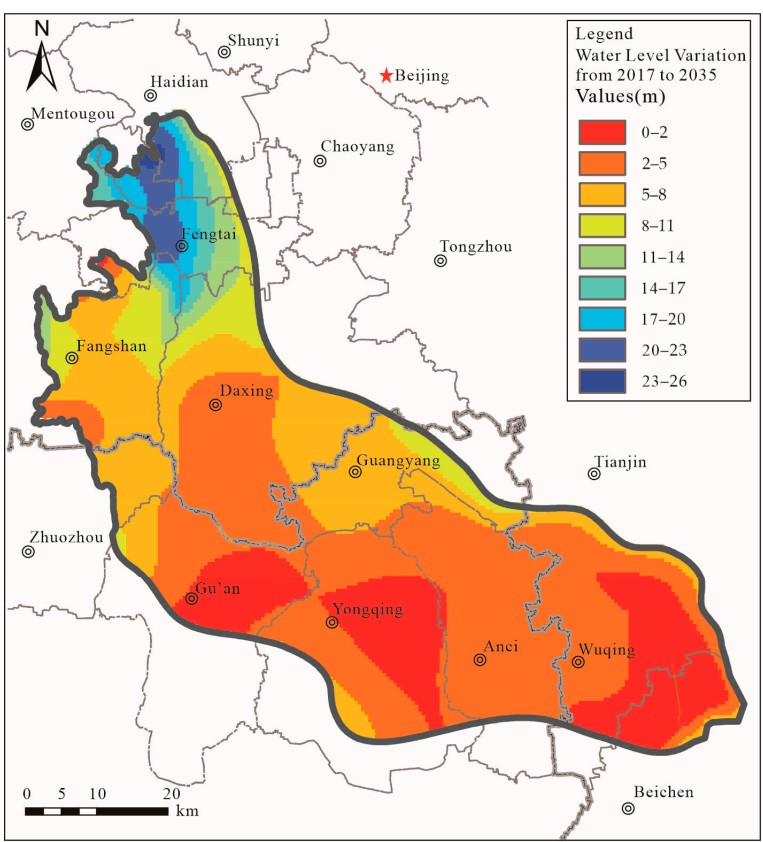

**Figure 10.** The predicted groundwater level variation of the study area.

## 5. Conclusions

In this paper, the fuzzy mathematics model, numerical simulation model, and random forest model have been linked to evaluate the effect of ecological water supplement in the Yongding River on groundwater restoration. The models established in this study have good performance and can reflect the features of groundwater flow. The infiltration volume of the ecological water supplement, the groundwater level variation, the impact range of the ecological water supplement, the contribution of ecological water supplement to the groundwater restoration, and the ecological water supplement suggestion have been systematically studied by using the linkage models.

The seepage rate of the upper reaches of the Yongding River could reach more than 0.7, and the downstream is generally about 0.2. The infiltration volume of the Yongding River channel accounts for about 60% of the total ecological water supplement, which can be effectively recharged into the groundwater aquifer. The contribution of ecological water supplement infiltration to the uplift of groundwater level is 25%, indicating that ecological water supplement could significantly promote the groundwater restoration in the study area.

The average groundwater level increase was about 5 m/year from 2018 to 2019. The groundwater recharge and discharge difference was changed from $-163 \times 10^6$ m$^3$ to $220 \times 10^6$ m$^3$ before and after ecological water supplement. Additionally, the infiltration of ecological water supplement could account for 21.41% of the contribution to the difference of groundwater budget.

By the end of 2019, the maximum distance affected by ecological groundwater supplement in the Yongding River was 8.13 km, and the impact range was 502.13 km$^2$. By 2035, under the scheme of increasing the ecological water supplement to $170 \times 10^6$ m$^3$/year and reducing $35.77 \times 10^6$ m$^3$/year groundwater exploitation in the downstream area, the entire groundwater level will be uplifted in the study area.

**Author Contributions:** Conceptualization, T.N. and W.C.; methodology, T.N.; software, T.N.; validation, T.N. and W.C.; formal analysis, T.N.; investigation, W.C.; resources, W.C.; data curation, W.C.; writing— original draft preparation, T.N.; writing—review and editing, T.N. and W.C.; supervision, W.C.; funding acquisition, W.C. All authors have read and agreed to the published version of the manuscript.

**Funding:** This research was funded by National Key R&D Program of China (2022YFC3703701), National Natural Science Foundation of China (41972262), and Hebei Natural Science Foundation for Excellent Young Scholars (D2020504032).

**Data Availability Statement:** Publicly available datasets were analyzed in this study. The precipitation data, evaporation data and ecological water supplement data can be found here: http://data.cma.cn/ and http://www.hwcc.gov.cn/wwgj/slsjk/ (accessed on 18 December 2022). The drilling data presented in this study are available on request from the corresponding author. The data are not publicly available due to the regulations of the Institute of Hydrogeology and Environmental Geology. Restrictions apply to the availability of groundwater level and exploitation data. Data was obtained from Hydrology Bureau of Haihe River Water Conservancy Commission and National Groundwater Monitoring Center. And are available from the authors with the permission of Hydrology Bureau of Haihe River Water Conservancy Commission and National Groundwater Monitoring Center.

**Acknowledgments:** This research was financially supported by National Key R&D Program of China (2022YFC3703701), National Natural Science Foundation of China (41972262), and Hebei Natural Science Foundation for Excellent Young Scholars (D2020504032). The authors would like to thank the reviewers and editors for their helpful comments and suggestions.

**Conflicts of Interest:** The authors declare no conflict of interest.

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
