# Peer review of "Effect of Ecological Water Supplement on Groundwater Restoration in the Yongding River Based on Multi-Model Linkage"

_water, doi:10.3390/w15020374_

Round 1

Reviewer 1 Report

For the future readers would be interesting to know if there are on not any possible sources of "artificial ecological water".  Without such kinds of explanations this work looks like formal and pure theoretical one. My advice is to include in manuscript these explanations.

Author Response

Appreciate for your kindness review of this article. And your comments on this paper is visionary.

We have  added the explainations of the potential ecological water supply sources  in this paper. Please see the attachment.

Thank you  for your helpful review and comments!

Reviewer 2 Report

The effect of ecological water supplement in river on groundwater restoration is not comprehensively understood yet. This manuscript studied the ecological water supplement on groundwater level variation in the Yongding River. The data and result are useful to understand the groundwater level changes by ecological water supplement. However, this manuscript is like a technical report, the mechanism of the ecological water supplement on groundwater restoration is not sufficiently discussed.
>>Abstract. The novel finding of this research should emphasized in the abstract.
>>The names of city in Figures 1(a), 3(a), 4 etc., are not the same as the names in text( e.g. lines 308 Fangshan and Daxing).
>>Section 4.1. The indicators in Table3 are not coincide with the driving factors in Figures 7 and 8.
>>Section 4. Discussion. The discussion in this manuscript is not well written and sufficiently discussed with references.

Author Response

Thank you very much for your carefully and paitient review of this paper!

The comments on this paper is  crucial and helpful!

We have revised this article according to your comments! Please see the attachment!

Thanks for your  helpful review!

Best Wishes!

Reviewer 3 Report

Dear authors, 

I had a pleasure to get Your manuscript for the review but I must express my concern about the quality level of Your work. Below I attach comments based from analysis.

P.3.l.98. km2 should be written in superscript

P.3. l.101-102. What is the base for the amount of precipitation and evaporation. Is it the total study area or a square meter? Please clarify and add.

Since the Yongding River Comprehensive Management and Ecological Restoration Project launched [44,45], the groundwater restoration effect should be reflected in the study area.

The sentence is confusing, what has been launched by???

p.3. l105. The regional groundwater transports….Do authors mean of groundwater flow??

…in the piedmont area.. Detailed explanation of this area separately is needed.

At some points authors use future, at some present and even past tense…

2.1. Hydrological Conditions in the Study Area does not cover strictly hydrological conditions. Instead it covers for geological and hydrogeological conditions. Either the title should be ajdusted or the content should be dispersed in different chapters in accordance to the content.

The aquifers mainly consist of coarse-grained sediments in the piedmont area, and gradually changed into fine er-grained sediments in the downstream plain (Figure 1b).  – Reference needed!

Therefore, the aquifer is mainly a single phreatic aquifer in the piedmont area, and gradually divided into multi-layer aquifers in the middle and downstream plain. - Reference needed!

The aquifer is mainly recharged by means of lateral runoff in the piedmont zone, precipitation infiltration and water infiltration of the Yongding River; it is discharged by means of artificial mining, evapora tion and lateral outflow of the aquifer. - Reference needed!

The bottom boundary of the Quaternary aquifer is impermeable sediment.- Reference needed!

Last paragraph of ch. 2.1. and Figure 2. does not fit the title of chapter 2.1.

2.2. Matertrials…shoud be Materials

Topographic data come from Grace satellite remote sensing inter pretation results.  - Reference needed!

Groundwater exploitation data are collected from Beijing, Tianjin and Hebei hydrological reports.  - Reference needed!

The data during the ecological restoration period from 2018 to 2019 is from Haihe River Water Conservancy Commission. - Reference needed!

The water level data are from monthly observations from national groundwater monitoring wells in the area from 2016 to 2019. - Reference needed!

The drilling data of the sections are derived from the hydrogeological survey of the Institute of Hydrogeology and Environmental Geology, Chinese Academy of Geological Sciences in 2019. - Reference needed!

The model was developed with MODFLOW. The grid size was 500 m × 500 m. The first layer was discretized into 15,850 effective cells, and the second layer was discretized into 11,653 effective cells.

What is the gurantee of grid convergence? This issue has to be diostinguished separately to detect possible sensitivity of model results to grid size.

p.5., l.191. …the shallow aquifer was mainly received ecological water supplement by infiltration…

What is the formulation of „was supplement“??

The calibration proceedure has to be shown in detals. It is far from enough to write: The model was fitted and calibrated by using the observation data of 64 typical observation wells in the study area from 2016 to 2019 and the measured groundwater flow field on 31 December 2019.

Hereby authors need to elaborate on:

-        If any BC was a part of calibration proceedure?

-        Which physical parameters were calibrated?

-        Which numerical parameters were calibrated?

No clue on how hydraulic conductivity, specific yield, and storage coefficient hve been determined. There is only one remark which lead towards conclusion hydraulic conductivirty was obtained via the calibration proceedure. I see this as a very relevant issue since the conductivity is the main factor controlling the seepage velocity (Darcy law). Furthermore I don't find anything about spatial variability of the hydraulic conductivity so one can assume homogeneity and isotropy of the aquifer. The latter is very strange to be found in nature.

Due to the hige number of mistakes in writing, plenty of issues manuscript lacks from and the way how the manuscript has been organised I suggest to update the manuscript, significantly improve its scientific contribution and prepsentation and do a ressubmittion.

Author Response

Thank you for your careful and exact review of this paper. This paper has been carefully revised and polished by a native English speaker. Please see the attachment.

Thank you for your suggestions!

Best Wishes!

Reviewer 4 Report

Dear authors, Its a good study but needs further improvement. Some specific comments are appended below.

Specific comments:

Line 2: The title may be modified.”  Research on the..”, is it required?

Line 16-17: The results obtained using the random forest model should be mentioned briefly in the abstract.

Line 23-26: These sentences may be improved to enhance clarity

Line 51: conceptual?

Line 54: based, not base

Line 88, 91: . And the? Can this be improved? Please also check elsewhere in the manuscript.

Line 103-105: Please check grammar; possibly, this sentence can be improved

Line 129: The section heading may be modified. Simple “Data collection” may be more appropriate

Line 237-242: Feature importance value distribution and Shapley value distribution should be elaborated a little more to facilitate easy understanding by the readers

Line 246: Infiltration volume would be more appropriate

Line 251: Table 1: Why seepage rate does not have a unit?

Line 286: which could be accurately reflected the variation…please check throughout the manuscript such grammar issues

Line 311: The groundwater level decreased mainly occurs?

Line 323-324: What is negative equilibrium state and positive equilibrium state?

Line 330: remove” the”, similarly in other places in the text

Line starting line 387: The discussion part is very poorly written. Hence, this section should be rewritten or merged with results section appropriately.

Author Response

Thank you for your careful and patient review of this article!

We have modified this paper. And a native English speaker has helped us to polish this article. Please see the attachment.

Thank you very much!

Best Wishes! 

Round 2

Reviewer 2 Report

The manuscript has been sufficiently improved as the reviewers' comments.

Reviewer 4 Report

Dear Authors,

There are still some issues with grammar and typological errors.

I would suggest a thorough check again.

Some are given below.

Line 2: remove “the”

Line 91: proposed an or a?

Line 268: “is not only provided” is wrong....please correct the sentence

Please check similar typological and grammatical mistakes errors throughout the manuscript.